# Insights on autophagosome–lysosome tethering from structural and biochemical characterization of human autophagy factor EPG5

Sung-Eun Nam[1,3], Yiu Wing Sunny Cheung [1,3], Thanh Ngoc Nguyen [2], Michael Gong[1], Samuel Chan[1], Michael Lazarou [2] & Calvin K. Yip [1✉]

Pivotal to the maintenance of cellular homeostasis, macroautophagy (hereafter autophagy) is an evolutionarily conserved degradation system that involves sequestration of cytoplasmic material into the double-membrane autophagosome and targeting of this transport vesicle to the lysosome/late endosome for degradation. EPG5 is a large-sized metazoan protein proposed to serve as a tethering factor to enforce autophagosome–lysosome/late endosome fusion specificity, and its deficiency causes a severe multisystem disorder known as Vici syndrome. Here, we show that human EPG5 (hEPG5) adopts an extended "shepherd's staff" architecture. We find that hEPG5 binds preferentially to members of the GABARAP subfamily of human ATG8 proteins critical to autophagosome–lysosome fusion. The hEPG5–GABARAPs interaction, which is mediated by tandem LIR motifs that exhibit differential affinities, is required for hEPG5 recruitment to mitochondria during PINK1/Parkin-dependent mitophagy. Lastly, we find that the Vici syndrome mutation Gln336Arg does not affect the hEPG5's overall stability nor its ability to engage in interaction with the GABARAPs. Collectively, results from our studies reveal new insights into how hEPG5 recognizes mature autophagosome and establish a platform for examining the molecular effects of Vici syndrome disease mutations on hEPG5.

[1] Life Sciences Institute, Department of Biochemistry and Molecular Biology, The University of British Columbia, Vancouver, BC, Canada. [2] Department of Biochemistry and Molecular Biology, Biomedicine Discovery Institute, Monash University, Melbourne, VIC, Australia. [3] These authors contributed equally: Sung-Eun Nam, Yiu Wing Sunny Cheung. ✉email: calvin.yip@ubc.ca

Macroautophagy (also known as autophagy) is the main pathway for degrading long-lived cytoplasmic macromolecules and full-sized organelles and represents a key component of the cellular homeostatic program. Under favorable growth conditions, basal autophagy serves as a quality control mechanism to selectively remove misfolded/aggregated proteins and dysfunctional organelles in the cytoplasm[1,2]. When cells encounter stress conditions, such as starvation, autophagy is upregulated to promote nonselective bulk degradation to generate basic building blocks to power essential metabolic reactions and to generate energy[3–5]. Because of autophagy's important roles in guarding normal cellular physiology, dysregulation of this degradation pathway is linked to many human pathologies ranging from neurodegeneration and cancer to infectious diseases[6,7]. An improved understanding of autophagy at the molecular level will generate insights into the basis of different human diseases and may reveal additional avenues for therapeutic intervention.

Autophagy degradation begins with the formation of a membrane precursor known as the phagophore. The phagophore expands in size, sequesters cytoplasmic materials, and self-seals to form a double-membrane transport vesicle called the autophagosome. The cargo-laden autophagosome then gets transported to and fuses with the lysosome or the late endosome, where the content of the autophagosome is ultimately digested by hydrolytic enzymes inside the lysosome[3–5,8]. The discovery of the ATG (autophagy related) genes by yeast genetic screening and the identification of the core autophagy machinery composed of 18 mostly conserved Atg proteins generated a framework for investigating the molecular mechanism of this multistep degradation pathway. Subsequent characterization of the core autophagy machinery consisting of five key functional groups (Atg1 kinase complex, autophagy-specific phosphatidylinositol 3-kinase/PI3K complex, Atg8 conjugation system, Atg12 conjugation system, Atg9 and Atg2–Atg18 complex) yielded mechanistic insights into autophagy initiation and autophagosome biogenesis, primarily in the yeast model system[3,9,10]. However, the mechanisms underlying the later steps of autophagy, including how autophagosome engages and ultimately fuses with the lysosome remain less well understood.

Recent studies in high eukaryotes have begun to unravel these mysteries. The identification of syntaxin 17 (STX17) and YKT6 as the autophagosomal "SNARE" that bind lysosomal VAMP8–SNAP29 and STX7–SNAP29, respectively, to form trans-SNARE complexes offered insights into the autophagosome–lysosome membrane fusion process[11,12]. Systematic gene deletion studies of the six homologs of human ATG8 (LC3A, LC3B, LC3C, GABARAP, GABARAPL1, and GABARAPL2) revealed that the three members of the GABARAP subfamily play critical roles to autophagosome–lysosome fusion[13]. Furthermore, the discovery of new non-ATG autophagy regulators, including the conserved multi-subunit HOPS complex, and the metazoan-specific proteins TECPR1, PLEKHM1, BRUCE, GRASP55, and EPG5 generated a growing list of additional proteins/protein complexes that participate in the terminal stage of autophagy[11,14–20]. However, the precise physiological functions of these newly identified autophagy factors, and exactly how they coordinate with one another and the GABARAP proteins to mediate autophagosome–lysosome fusion are not fully understood.

Originally discovered in the *Caenorhabditis elegans* genetic screen for metazoan autophagy genes, EPG5 (ectopic P-granules autophagy protein 5) is a large-sized (~292 kDa) protein proposed to regulate fusion specificity between autophagosomes and lysosomes. Notably, epg-5 deficiency in *C. elegans* causes nonspecific fusion of autophagosomes with other endocytic vesicles and the formation of abnormally large non-degradative vesicles[20,21]. Subsequent studies in *C. elegans* and human cell

lines showed that EPG5 is recruited to the lysosome/late endosome by the small GTPase RAB7, and EPG5 has the ability to bind the autophagosome surface protein and ATG8 homolog human LC3B or *C. elegans* LGG-1 via two LC3-interacting region (LIR) motifs composed of a conserved sequence [(W/F/Y)-$X_1$-$X_2$-(I/L/V)][20]. Together with the finding that *C. elegans* EPG-5 is capable of stabilizing and facilitating assembly of the STX17–SNAP29–VAMP8 trans-SNARE complex in vitro, these new data led to the model that EPG5 functions as an autophagy tethering factor that mediates initial interaction between the autophagosome and the lysosome[20].

At around the time when EPG5's role in autophagy was uncovered, clinical genetics analysis revealed that recessive mutations of the gene encoding human EPG5 (hEPG5) cause Vici syndrome, a rare but severe multisystem disorder characterized by agenesis of the corpus callosum, cataracts, cardiomyopathy, hypopigmentation, and combined immunodeficiency[22–24]. Approximately 100 cases of Vici syndrome have been reported to date with a median survival time of 24 months[22–24]. Analyses of primary cells isolated from patients showed an accumulation of autophagosomes attributed to deficiency in autophagosome–lysosome fusion[24,25]. Interestingly, epg5[−/−] knockout mice exhibit neurodegenerative features resembling human amyotrophic lateral sclerosis[26]. Although many Vici syndrome mutations have been mapped, the effects of these mutations on EPG5's structure and function are not known.

By developing a method to produce recombinant full-length hEPG5, we were able to comprehensively characterize the structural and biochemical properties of this large-sized putative autophagy tethering factor. We found that hEPG5 adopts an extended architecture reminiscent to tethering factors found in other membrane trafficking pathways. We also found that hEPG5 shows preferential binding to the GABARAP subfamily of ATG8 proteins, and this interaction involves a complex interplay between the two LIR motifs exhibiting differential binding affinities. We further showed that hEPG5–GABARAP interaction is required for hEPG5 recruitment to mitochondria during PINK1/Parkin-mediated mitophagy. Lastly, the common recurrent Vici syndrome mutation Q336R did not affect the overall architecture, stability, and GABARAP-binding ability of hEPG5.

## Results

**hEPG5 adopts an extended overall architecture.** With 2579 amino acid residues and an overall molecular mass of ~290 kDa, hEPG5 is one of the largest regulators in the autophagy pathway identified to date. Due in part to technical challenges associated with purification of this large-sized protein, nothing is currently known about the structural properties of hEPG5. To overcome this barrier, we developed a baculovirus-insect cell-based system to overexpress the recombinant full-length hEPG5 and an anti-FLAG affinity chromatography coupled with glycerol density ultracentrifugation approach to purify the recombinant protein. This procedure enabled us to obtain highly purified hEPG5 suitable for biochemical and structural characterization (Fig. 1a). Analytical gel filtration chromatography of purified hEPG5 showed that it elutes at a volume corresponding to a predicted molecular weight higher than its calculated mass (Fig. 1b). This suggests that hEPG5 either exists as an obligate oligomer or adopts a non-globular overall shape. We next examined hEPG5 by negative stain single-particle electron microscopy (EM). Raw images not only revealed highly elongated particles with a distinct curvature at one end, but also showed that hEPG5 is monomeric (Fig. 1c). Two-dimensional (2D) analysis emphasized that hEPG5 has an overall architecture resembling a "shepherd's staff" and composed of a rigid round

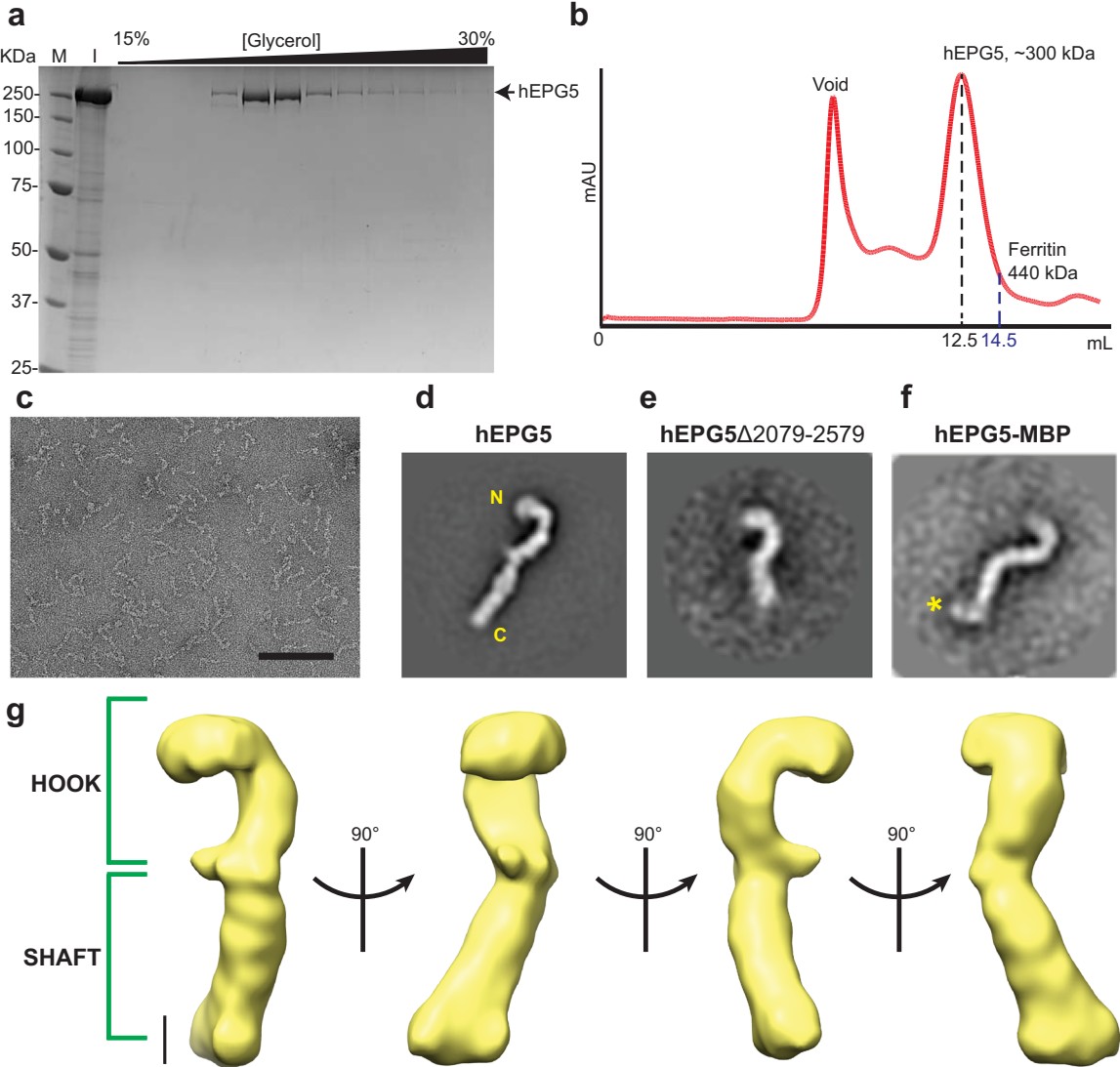

**Fig. 1 Overall architecture of hEPG5. a** SDS–PAGE of His-FLAG-hEPG5 glycerol gradient ultracentrifugation fractions, stained with Coomassie Blue. M and I represent the protein marker and input, respectively. **b** Analytical gel filtration chromatography of His-FLAG-hEPG5. Elution volume of hEPG5 (~300 kDa including the tags) is indicated by black dashed line and the molecular weight standard ferritin (440 kDa) is indicated by blue dashed line. **c** A representative raw image of negatively stained hEPG5 (Scale bar: 100 nm). **d** Representative 2D class averages of wild-type hEPG5, with the location of N- and C- termini indicated in yellow. **e** Representative 2D class averages of hEPG5$^{\Delta2079-2579}$. **f** Representative 2D class averages of C-terminal maltose-binding protein (MBP)-tagged hEPG5. MBP density is shown by asterisk (yellow). **g** 3D reconstruction of hEPG5 revealing the hook and shaft regions (Scale bar: 5 nm).

"hook" connected to an extended and more flexible "shaft" (Fig. 1d). A "thumb"-shaped protrusion is also present between the hook and the shaft. The length of hEPG5 is estimated to be ~ 375 Å, a value consistent with the maximum dimensions observed for different tethering complexes found in conventional membrane trafficking pathway[27,28]. A three-dimensional (3D) reconstruction calculated from the negative stain EM data showed that hEPG5 is nonplanar with its two "ends" projected toward opposite directions (Fig. 1g). To determine which regions of hEPG5 adopts the two prominent substructures, we generated and purified a truncated version of hEPG5 lacking the C-terminal 500 residues (designated hEPG5$^{\Delta2079-2579}$). 2D negative stain EM analysis showed that this hEPG5 truncation mutant, while adopting an overall architecture reminiscent of full-length hEPG5, contains a shorter shaft, indicating that the C-terminus of hEPG5 is located at the tip of the shaft (Fig. 1e). We also generated a fusion construct with maltose-binding protein (MBP)

fused to the C-terminus of hEPG5 and purified this fusion protein for negative stain EM. In agreement with our termini assignment from deletion analysis, 2D EM analysis showed an extra density projected from the tip of the shaft (Fig. 1f).

**hEPG5 interacts preferably with the GABARAP subfamily of ATG8 proteins.** hEPG5's extended architecture seems well suited to its proposed role in tethering the autophagosome to the lysosome prior to fusion. Tethering factors mediate longer range interaction between the transport vesicle and its target organelle[28,29]. The recent finding that hEPG5 is capable of binding LC3B indicated that hEPG5 likely recognizes the autophagosome via this human ATG8 protein, which localizes to both the inner and outer membrane of the autophagosome[20]. However, there are six homologs of ATG8 proteins in humans (LC3A, LC3B, LC3C, GABARAP, GABARAPL1, and

GABARAPL2)[30,31] and it is unclear if hEPG5 is capable of binding other members of the ATG8 family. We therefore subjected purified FLAG-tagged full-length hEPG5 to a systematic GST (glutathione S-transferase) pull-down analysis involving all six GST-tagged human ATG8 homologs. Our fluorescence-based western blotting showed that the three GABARAP subfamily of ATG8 proteins (GABARAP, GABARAPL1, and GABARAPL2) precipitated 2.5 times more hEPG5 compared to the three members of the LC3 subfamily (LC3A, LC3B, and LC3C), indicating that hEPG5 binds preferentially to the GABARAP's (Fig. 2a, b). Interestingly, hEPG5 has different affinities toward the three LC3 subfamily members, with the strongest interaction with LC3C and the weakest with LC3B.

**hEPG5 binds GABARAP and other ATG8 proteins via a tandem LIR motif.** ATG8 proteins typically bind the so called LIR motifs of their cognate binding partner[31,32]. Although previous studies have shown that two tandemly arranged motifs between residues 550 and 570 of hEPG5 ($^{550}$WTLV$^{553}$ and $^{567}$WILL$^{570}$) are essential for interaction with LC3B, other putative LIR motifs are also predicted along the entire length of hEPG5[20]. To find out which of these putative LIR's of hEPG5 is/are responsible for mediating the high-affinity interaction with the GABARAP proteins, we first applied a deletion mapping approach that involves purifying hEPG5 truncation mutants and assessing their abilities to bind GABARAP by GST pulldown. Out of the series of truncation mutant constructs we designed and generated, only three could be purified at sufficient levels for biochemical analyses (Supplementary Fig. 1a, b). These include hEPG5$^{\Delta 1-548}$ which is devoid of the region between the N-terminus and the two previously characterized LIR motifs, hEPG5$^{\Delta 1-1198}$ which excludes the entire N-terminal region, and hEPG5$^{\Delta 1770-2579}$ which excludes the entire C-terminal region, but shares a high degree of sequence identity with *C. elegans* EPG-5, which is approximately half the size of hEPG5 and consists of 1599 amino acid residues. Our pull-down results showed that hEPG5$^{\Delta 1770-2579}$ binds GABARAP equally well compared to wild-type hEPG5, suggesting that the entire C-terminal region of hEPG5 is dispensable to GABARAP interaction (Fig. 2c). The observation that hEPG5$^{\Delta 1-548}$, but not hEPG5$^{\Delta 1-1198}$ was pulled down by GABARAP suggested that the GABARAP-binding site is located between residues 548 and 1198 of hEPG5. We next mixed recombinant hEPG5 with GST-tagged GABARAP, purified the hEPG5–GABARAP complex, and examined the purified complex by negative stain EM. Our 2D analysis revealed an extra density present along the N-terminal hook shape structure and supported by the protruding "thumb" (Fig. 2d), confirming the general location of the GABARAP-binding domain from our deletion mapping experiment. Interestingly, we found that hEPG5$^{\Delta 1-548}$ was precipitated by GABARAP at a higher level compared to wild-type hEPG5. This observation could be attributed to increased accessibility of one or more LIR motifs upon removal of structural elements located in the N-terminal region of hEPG5 (Fig. 2c).

To determine which of the three putative LIR motifs between residues 548 and 1198 ($^{550}$WTLV$^{553}$, $^{567}$WILL$^{570}$, and $^{794}$FIKI$^{797}$) are required for GABARAP interaction, we first generated three hEPG5 mutants in which the key first aromatic residue of each of the three LIR motifs was replaced by an alanine (Supplementary Fig. 1c–e), and then assessed the ability of these mutants to bind GABARAP by GST pulldown. We found that mutations to the first two LIR motifs in this region (W550A and W567A) severely or mildly diminish hEPG5's interaction with GABARAP, respectively, underscoring their importance in binding GABARAP (Fig. 2e and Supplementary Fig. 2a, b). By

contrast, hEPG5$^{F794A}$ which contains mutation to the third LIR motif binds GABARAP as strongly as wild-type hEPG5, suggesting that this LIR motif is not required for GABARAP interaction (Supplementary Fig. 2c). We also generated the W550A/W567A double mutant (Supplementary Fig. 1f) and showed that it completely abolished GABARAP binding (Fig. 2e and Supplementary Fig. 2d). Collectively, these results indicated that the tandem LIR motifs, previously shown to bind LC3B[20], mediate high-affinity interaction with GABARAP. Furthermore, the first LIR (hereafter denoted LIR1) appears to play a more dominant role than the second LIR (hereafter denoted LIR2) in this interaction.

**LIR2 peptide shows higher binding affinity toward GABARAP proteins than LIR1 peptide.** To better understand how LIR1 and LIR2 work in conjunction with one another to mediate high-affinity interaction with GABARAP, we first decoupled the two LIR motifs by synthesizing peptides corresponding to LIR1 ($^{546}$GSGTWTLVDEG$^{556}$) or LIR2 ($^{560}$DEDPETSWILLN$^{571}$), and used isothermal titration calorimetry (ITC) to measure the binding constants of these two peptides with GABARAP and other human ATG8 proteins. For LIR1, we found that, in agreement with our pull-down data with full-length hEPG5, this peptide shows higher affinity toward all three GABARAP subfamily members compared to the three LC3 subfamily members (Fig. 3a and Supplementary Table 1). Notably, the strongest binding was observed for GABARAP ($K_d$ of 7.47 μM), followed by GABARAPL1 and GABARAPL2 (8.54 μM and 11.79 μM, respectively). By contrast, the three LC3 subfamily members bind weakly, and we could only accurately determine the $K_d$ of LIR1 with LC3A, which is three times higher than that for LIR1-GABARAP.

For LIR2, our ITC experiments revealed that although this peptide retains strong preference toward the three GABARAP subfamily members, it exhibits substantially higher affinity toward all six ATG8 homologs (Fig. 3b and Supplementary Table 1). More specifically, the $K_d$ value of LIR2 with GABARAP (0.16 μM), GABARAPL1 (0.09 μM), and GABARAPL2 (0.68 μM) are ~45-fold, 100-fold, and 15-fold lower than those measured for LIR1, respectively. Similarly, the $K_d$ for LIR2 with LC3A and LC3B (1.75 μM and 4.07 μM, respectively) are approximately ten times lower than that determined for LIR1. The observation that LIR2 binds more tightly to GABARAP and other ATG8 homologs than LIR1 was unexpected, given that our pull-down analysis on the full-length hEPG5 LIR mutants indicated that LIR1 plays a more dominant role in this interaction. This discrepancy could be explained by the relative inaccessibility of LIR2 in the context of full-length hEPG5 prior to LIR1 contacting GABARAP and possibility causing a local conformational change.

**LIR2 binds canonical binding site on GABARAPL1.** We next examined how LIR2 mediates high-affinity interaction with the GABARAP subfamily proteins by co-crystallizing LIR2 in complex with GABARAPL1 and determining the crystal structure of this complex at 1.91 Å resolution (Table 1). There are two copies of LIR2–GABARAPL1 present in the asymmetric unit and their overall structures are essentially identical to one another (Fig. 4a and Supplementary Fig. 3a, b). Our crystal structure revealed that LIR2 binds GABARAPL1 at the canonical LIR-binding site through a network of hydrophobic and electrostatic interactions. The critical aromatic residue W567$^{hEPG5}$ is inserted into hydrophobic pocket 1 (HP1) of GABARAPL1, whereas the hydrophobic residue L570$^{hEPG5}$ is inserted into hydrophobic pocket 2 (HP2; Fig. 4b). Within HP1, side chains of the residues P30, L50, and F104 of GABARAPL1 form hydrophobic interaction with

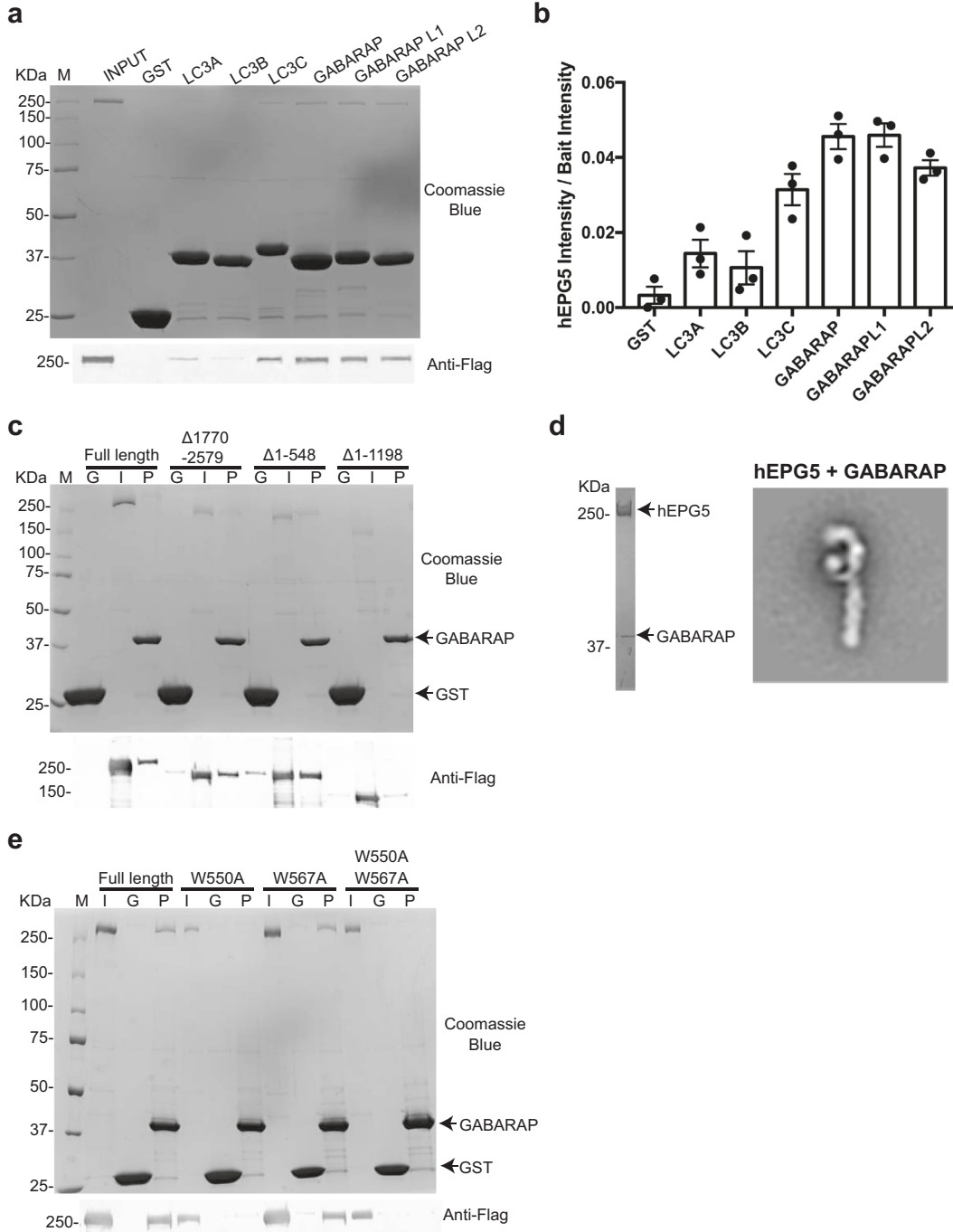

**Fig. 2 In vitro pull-down assays of hEPG5 and the LC3/GABARAP subfamily. a, c, e** Precipitation of FLAG-tagged wild-type and mutant hEPG5 using GST (control) and N-terminally GST-tagged LC3/GABARAP subfamily proteins as baits. Representative SDS–PAGE stained with Coomassie Blue (top panel) shows the input. Representative western blot (bottom panel) was probed by anti-FLAG antibodies. M, G, I, and P represent the protein marker, GST control, hEPG5 input, and pulldown, respectively. Experiments were performed in triplicates. **a** In vitro pull-down assays of hEPG5 with GST (control) and N-terminally GST-tagged LC3/GABARAP subfamily proteins used as baits. **b** Quantification of hEPG5 binding to GST (control) and GST-LC3/GABARAP subfamily proteins. Data are shown as mean ± SEM of three individual experiments. **c** Full-length and truncated hEPG5 (hEPG5$^{\Delta1770-2579}$, hEPG5$^{\Delta1-548}$, and hEPG5$^{\Delta1-1198}$) pull-down analysis using GST-GABARAP as bait. **d** (Left panel) SDS–PAGE of His-FLAG-hEPG5 in complex with GST-GABARAP isolated by nickel-immobilized metal affinity chromatography, stained with Coomassie Blue. (Right panel) Representative 2D class averages of hEPG5 in complex with GST-GABARAP. The additional GST-GABARAP density is denoted by the white arrow. **e** Wild-type and hEPG5 mutant (hEPG5$^{W550A}$, hEPG5$^{W567A}$, and hEPG5$^{W550A/W567A}$) pull-down analysis using GST-GABARAP as bait.

W567$^{hEPG5}$ (Fig. 4c). On the other hand, side chains of the residues lining HP2 (Y49, V51, F60, L63, and I64) are engaged in hydrophobic contacts with L570$^{hEPG5}$ (Fig. 4f). In addition, W567$^{hEPG5}$ forms electrostatic interaction with the carboxyl group on E17$^{GABARAPL1}$ side chain at HP1, and the main chain

carbonyl oxygen and NH group of L570$^{hEPG5}$ forms hydrogen bonds with the guanidinium group of R28$^{GABARAPL1}$ and carbonyl oxygen of L50$^{GABARAPL1}$ at HP2 (Fig. 4c, f). The central residues of LIR2 also contributed to GABARAPL1 binding. I568$^{hEPG5}$ side chain forms hydrophobic interaction with the

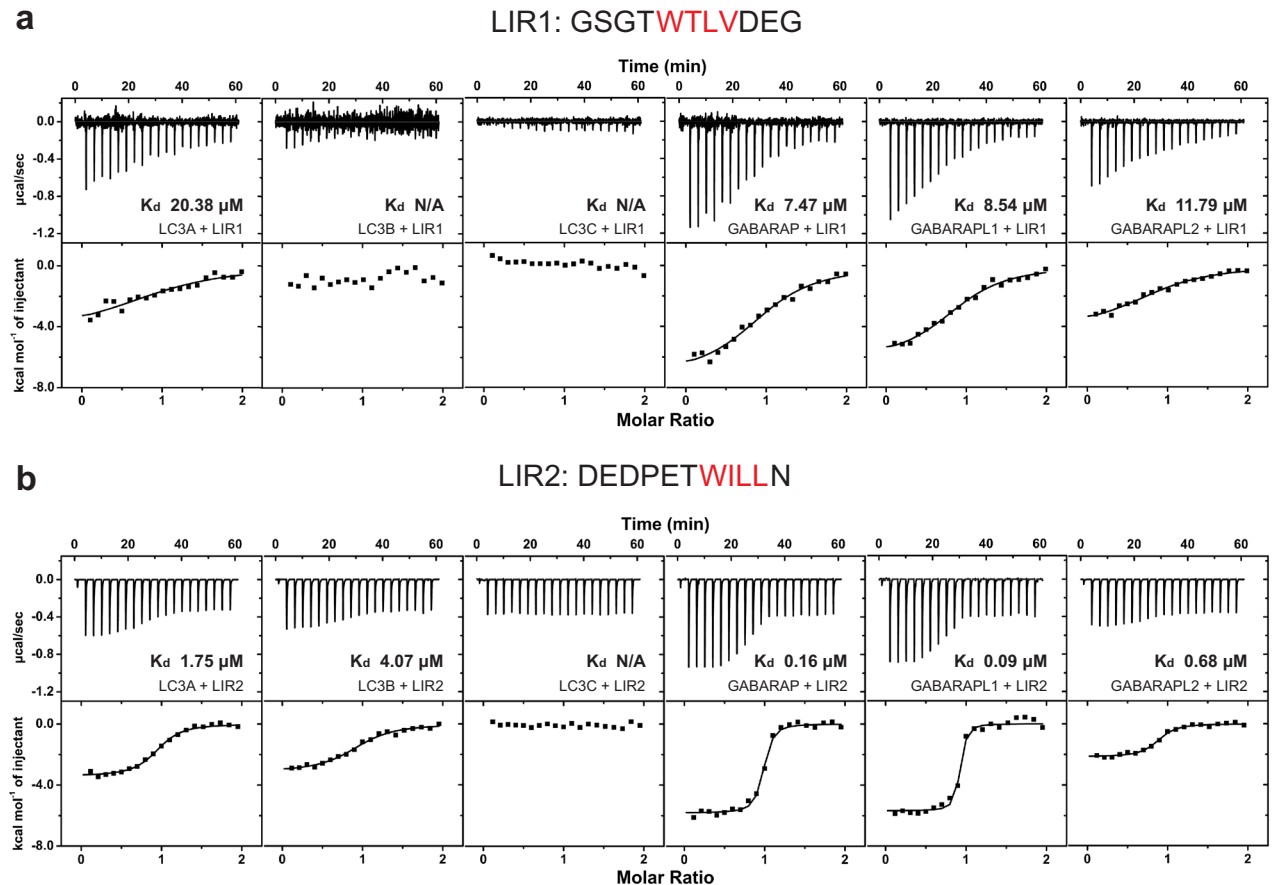

**Fig. 3 Isothermal titration calorimetry analysis of hEPG5 LIR motifs with LC3/GABARAP subfamily proteins. a, b** Isothermal titration calorimetry (ITC) results of LIR1 peptide ([546]GSGT<u>WTLV</u>DEG[556]) (600 μM) (**a**), and LIR2 peptide ([560]DEDPETS<u>WILL</u>N[571]) (600 μM) (**b**) titrating into individual LC3/GABARAP subfamily proteins (60 μM). The top diagram in each ITC plot shows the raw data and the bottom diagram shows the integrated data.

### Table 1 Data collection and refinement statistics.

| | GABARAPL1 + hEPG5-LIR2 complex |
|---|---|
| *Data collection* | |
| Space group | C121 |
| Cell dimensions | |
| $a, b, c$ (Å) | 122.95, 33.08, 78.86 |
| $\alpha, \beta, \gamma$ (°) | 90.00, 114.29, 90.00 |
| Resolution (Å) | 57.00–1.91 (1.91) |
| $R_{sym}$ or $R_{merge}$ | 0.147 (0.703) |
| $I/\sigma I$ | 4.60 (1.69) |
| Completeness (%) | 92.36 (60.25) |
| Redundancy | 3.2 (2.9) |
| *Refinement* | |
| Resolution (Å) | 57.00–1.91 (1.91) |
| No. reflections | 68435 |
| $R_{work}$ / $R_{free}$ | 0.1969/0.2448 |
| No. atoms | 2353 |
| Protein | 2097 |
| Ligand/ion | 15 |
| Water | 241 |
| *B*-factors | 25.45 |
| Protein | 24.64 |
| Ligand/ion | 48.68 |
| Water | 31.03 |
| R.m.s.ds | |
| Bond lengths (Å) | 0.013 |
| Bond angles (°) | 1.11 |

Values in parentheses are for highest-resolution shell.

aromatic side chain of Y49[GABARAPL1], and NH group and carbonyl oxygen of I568[hEPG5] forms hydrogen bonds with the main chains of K48[GABARAPL1] and L50[GABARAPL1] (Fig. 4d). L569[hEPG5] forms hydrophobic interaction with the side chain of Y25[GABARAPL1] and L50[GABARAPL1] (Fig. 4e). hEPG5-LIR2 engages in interaction with GABARAPL1 in a very similar fashion as other GABARAPL1 binding partners (PDB:5DPT[33]; PDB:5LXI[34]; PDB:5YIP[35]; PDB:6HOL[36]; and PDB:6HOI[36]), with root-mean-square deviation (r.m.s.d.) of these structures ranging from 0.65 to 1.00 Å (Supplementary Fig. 3c).

The acidic cluster at the N-terminal of LIR motif has been previously shown to be important for LC3/GABARAP proteins interaction[31,36–45]. Although our LIR2 peptide encodes the N-terminal acidic cluster ([560]DED[562]), these residues are not visible in the electron density map, suggesting that this cluster does not play an important role in mediating GABARAPL1 binding. Instead, hydrogen bonding between GABARAPL1 and the three preceding residues of LIR ([564]ETS[566]), was observed: (1) carboxyl group on E564[hEPG5] side chain with the hydroxyl group on Y25[GABARAPL1] side chain; (2) the main chain of T565[hEPG5] with the ε-amino group on K46[GABARAPL1] side chain; and (3) the hydroxyl group on S566[hEPG5] side chain with the ε-amino group on K48[GABARAPL1] side chain (Fig. 4g), which was previously observed in PIK3C3–LIR–GABARAP structure[36]. The C-terminal residue of LIR2, N571, also forms hydrogen bonds with the guanidinium group on R28[GABARAPL1] (Fig. 4h).

We then compared our LIR2–GABARAPL1 crystal structure with the previously reported apo-GABARAPL1 crystal structure (PDB:2R2Q). We found that the side chain of K46[GABARAPL1]

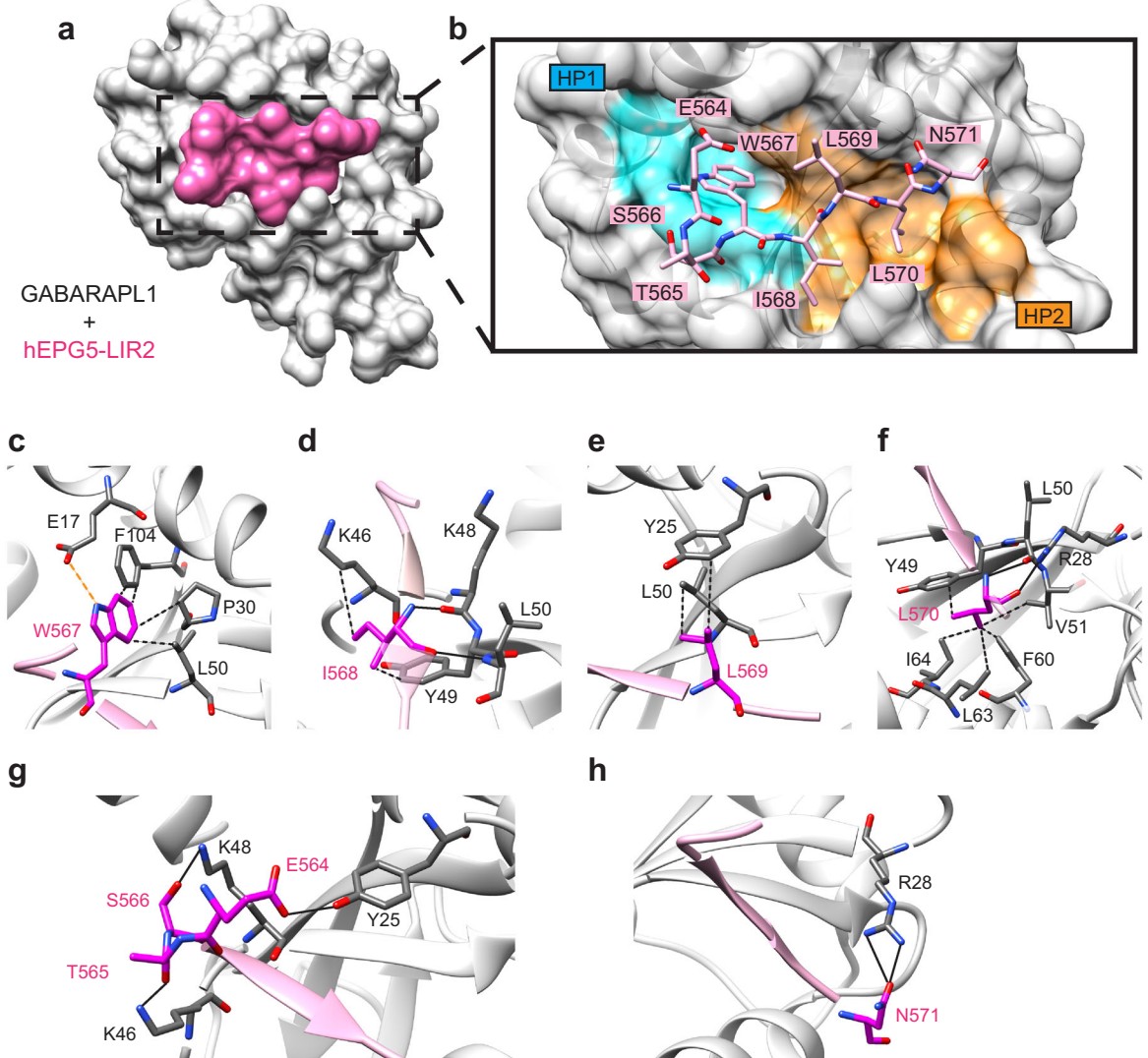

**Fig. 4 Crystal structure of the LIR2–GABARAPL1 complex. a** Surface representation of complex between LIR2 peptide (magenta) and GABARAPL1 (gray) at 1.91 Å. **b** Close-up view of LIR2 peptide (sticks representation in pink) binding to canonical binding site on GABARAPL1 (ribbon and transparent surface representation in gray). Residues Trp567 and Leu570 of LIR2 inserted into hydrophobic pocket 1 (HP1 in cyan) and hydrophobic pocket 2 (HP2 in orange) through hydrophobic interaction, respectively. **c–f** Close-up view of the interactions between each of the LIR2 motif residues (magenta) and residues on GABARAPL1 (dim gray). Black and orange dotted lines represent hydrophobic and electrostatic interaction, respectively; black solid lines represent hydrogen bonds. **c** Side chain of Trp567 interacts with side chain of Pro30, Leu50, and Phe104 through hydrophobic interaction, as well as the carboxyl group on Glu17 side chain through electrostatic interaction. **d** Side chain of Ile568 interacts with side chain of Lys46 and Tyr49 through hydrophobic interaction; main chain of Ile568 forms hydrogen bonds with main chain of Lys48 and Leu50. **e** Side chain of Leu569 interacts with side chain of Tyr25 and Leu50 through hydrophobic interaction. **f** Side chain of Leu570 interacts with side chain of Tyr49, Val51, Phe60, Leu63, and Ile64 through hydrophobic interaction; main chain of Leu570 interacts with main chain of Leu50 and guanidinium group on Arg28 side chain. **g**, **h** Close-up view of the interactions between each of the LIR2 N- and C-terminal residues (magenta) and residues on GABARAPL1 (dim gray). Solid lines represent hydrogen bonds. **g** LIR2 N-terminal residues Glu564 side chain, Thr565 main chain, and Ser566 side chain form hydrogen bonds with side chain of Tyr25, Lys46, and Lys48, respectively. **h** LIR2 C-terminal residue Asn571 side chain form hydrogen bonds with guanidinium group on Arg28 side chain.

undergoes a conformational rearrangement upon LIR2 binding (Supplementary Fig. 3d–f). This lysine conformational rearrangement has previously been shown to be important for LIR motif binding in LC3 subfamily proteins, as well as GABARAP and GABARAPL2, suggesting that this mechanism is conserved amongst mammalian LC3/GABARAP proteins (K49 for LC3A/B, K55 for LC3C, and K46 for GABARAP/L1/L2)[46–48]. Within apo-GABARAPL1, the side chain of K46 forms hydrogen bond with the main chain of K48, as well as hydrophobic interaction with the aromatic ring of the Y49. Upon LIR2 peptide binding, such interactions are disrupted, and the side chain of K46$^{\text{GABARAPL1}}$ shifts outward by 8.0 Å. This creates space to accommodate

I568$^{\text{hEPG5}}$ of LIR2 to bind and engage in hydrophobic interaction with K46$^{\text{GABARAPL1}}$ and Y49$^{\text{GABARAPL1}}$, including a hydrogen bond with the main chain of K48$^{\text{GABARAPL1}}$, as described above (Fig. 4d).

**hEPG5 requires GABARAP to localize to mitochondria during PINK1/Parkin-mediated mitophagy.** The high-affinity interaction between hEPG5 and GABARAP's made us contemplate if hEPG5 functions by first recognizing mature autophagosome at its site of formation before being trafficked together to a location where autophagosome–lysosome/late endosome fusion takes

place. Although previous studies showed that hEPG5 localizes to the perinuclear region, as well as diffusely in the cytoplasm in basal conditions[20], it is unclear if the GABARAP's has a role in this localization pattern. We therefore transfected hEPG5-GFP into four different HeLa cell lines: wild type, LC3-TKO which contains deletion of all three genes encoding the LC3 subfamily members, GABARAP-TKO which contains deletion of all three genes encoding the three GABARAP subfamily members, and ATG8-hexaKO which contains deletion of all six genes encoding the six human ATG8 homologs, and examined hEPG5's localization by confocal microscopy. We found that hEPG5-GFP shows the same localization pattern in all four different cell lines (Fig. 5a), indicating that this tethering factor traffics to the lysosome/late endosome independent of the GABARAP's.

To further delineate the role of hEPG5–GABARAP interaction in autophagy, we next analyzed hEPG5's localization under mitophagy-inducing condition. We used the well-established approach of activating PINK1/Parkin-dependent mitophagy by treating cells with oligomycin and antimycin A[49]. Upon induction of mitophagy, hEPG5-GFP localizes to punctate structures on or next to mitochondria in WT and LC3-TKO cells (Fig. 5b). Examination of hEPG5-GFP in GABARAP-TKO and ATG8-hexaKO showed that the absence of the GABARAP's appears to prevent the formation of these structures. These results indicate that hEPG5 requires the GABARAP's for recruitment to mitochondria during PINK1/Parkin-dependent mitophagy, and that hEPG5 functions downstream of the GABARAP's to drive autophagosome–lysosome/late endosome fusion.

**Vici syndrome mutation Q336R does not affect structural integrity and stability of hEPG5.** With a robust system to produce recombinant hEPG5 in place, we utilized this platform to more thoroughly examine the effects of Vici syndrome mutations on the structural and biochemical properties of hEPG5. A common missense mutation discovered from studies on two large cohorts of Vici syndrome is a nucleotide mutation at position 1007 of the *epg5* gene. This mutation results in single residue change (Gln336Arg; Q336R) in hEPG5 protein[22,24,50–52]. We decided to first focus on this disease mutation and examine its effect on the hEPG5 protein. We were able to express and purify hEPG5$^{Q336R}$ at similar yield compared to wild-type hEPG5 (Fig. 1a and Supplementary Fig. 1g). Negative stain EM analysis on hEPG5$^{Q336R}$ showed no change in overall architecture and subunit stoichiometry compared to that of wild-type hEPG5 (Fig. 6a). We next used the thermal shift assay to assess the stability of the mutant protein and found that hEPG5$^{Q336R}$ is slightly more stable than wild-type hEPG5, with an estimated 1.5 °C higher melting temperature (Fig. 6b, c). Lastly, GST pull-down assay shows that the Q336R mutation did not affect hEPG5's ability to bind to GABARAP and other human ATG8 proteins (Fig. 6d).

## Discussion

For all intracellular trafficking pathways, including autophagy, a transport vesicle must fuse specifically with its target organelle to ensure each cargo can reach and be delivered to its correct destination[53,54]. Tethering factors are a diverse family of proteins and protein complexes that play critical roles in defining and enforcing this specificity through mediating initial engagement between a transport vesicle and its target and facilitating the fusion event[54,55]. Recent studies on *C. elegans* EPG-5 by the Zhang group led to the proposal that this large-sized protein serves as an autophagy tethering factor, as it possesses two features found in tethering factors of other membrane trafficking pathways: (1) the ability to bind the transport vesicle (autophagosome via LC3B) and the target organelle (late endosome/

lysosome via RAB7), and (2) the ability to facilitate the formation of the STX17–SNAP29–VAMP7/8 trans-SNARE complex that mediates membrane fusion[20]. The first structural information on full-length hEPG5 reported here further substantiated this hypothesis by showing that hEPG5 adopts a relatively elongated overall shape, an architecture reminiscent of the "appendages" substructures found in multi-subunit tethering complexes, including COG[56], TRAPPII[57], exocyst[58], and HOPS[59]. While our 2D and 3D EM analyses suggested that hEPG5 is relatively rigid, its C-terminal "shaft" exhibits conformationally flexibility as has been observed for most tethering factors characterized to date. Lastly, the overall length of hEPG5 matches closely to the longest dimension of the tethering complexes, which was thought to be evolved to mediate interactions at distance beyond that of the SNARE fusogen[28,29,60,61]. hEPG5 is predicted to be composed of predominantly helical structures based on several different secondary structure prediction algorithms. Future high-resolution structural analysis of hEPG5 will determine if the extended substructures of hEPG5 are constructed by helical bundles that are arranged in a similar fashion as those observed in high-resolution crystal structures of tethering complexes subunits.

Previous multigene deletion studies of the six human ATG8 homologs demonstrated that the three GABARAP subfamily members play crucial roles in autophagosome–lysosome/late endosome fusion[13]. In agreement with this earlier finding, we observed that full-length hEPG5 shows a strong preferential binding to the three GABARAP proteins. We also demonstrated the importance of hEPG5–GABARAP interaction in autophagy by demonstrating that hEPG5 requires GABARAP's to localize to mitochondria in PINK1/Parkin-dependent mitophagy. Although multiple putative LIR motifs are predicted along the entire length of hEPG5, only the two tandemly arranged LIR motifs previously shown to mediate LC3B interaction are directly involved in binding GABARAP. Interestingly, the sequences of these two LIR motifs do not resemble the recently characterized GABARAP interaction motif or GIM ([W/F]-[V/I]-X$_2$-V)[33], indicating that other structural or biochemical features on hEPG5 contribute to its preference for GABARAP.

Our finding that the tandem LIR motifs (LIR1 and LIR2) are both essential for optimal binding to GABARAP raises the question as to why two motifs were evolved. In an attempt to understand the functional relationship between the two LIR motifs, we observed that although LIR1 clearly shows a more dominant role than LIR2 in mediating GABARAP interaction, the isolated LIR2 binds GABARAP with substantially higher affinity. These seemingly contradictory results, though initially perplexing, suggested that a more complex relationship exists between the two LIR motifs. Our observation that deletion of the N-terminal region of hEPG5 can alleviate potential inhibitory effects on GABARAP binding indicate that one or both of these LIR motifs might be inaccessible in the context of full-length hEPG5. Based on this result, we proposed a "two-factor authentication" step-wise binding model, in which LIR1 serves as an anchoring motif which makes initial contact with GABARAP, possibly at a noncanonical site. This binding would trigger a local conformational change making LIR2 accessible to binding with the canonical site on GABARAP (Fig. 7a). The ability for a tandem LIR motifs to bind simultaneously to a single ATG8 protein has been previously observed for RavZ, a *Legionella pneumophila* effector protein that inhibits xenophagy by cleaving lipidated ATG8 proteins, such as LC3B. Crystallographic analysis of the N-terminal tandem LIR motifs of RavZ in complex with LC3B revealed that the tandem motifs adopt a novel beta-sheet conformation with the second LIR binding in a noncanonical fashion[45]. As we were unable to obtain well-ordered crystals of hEPG5-LIR1 in complex with GABARAP proteins, likely due to

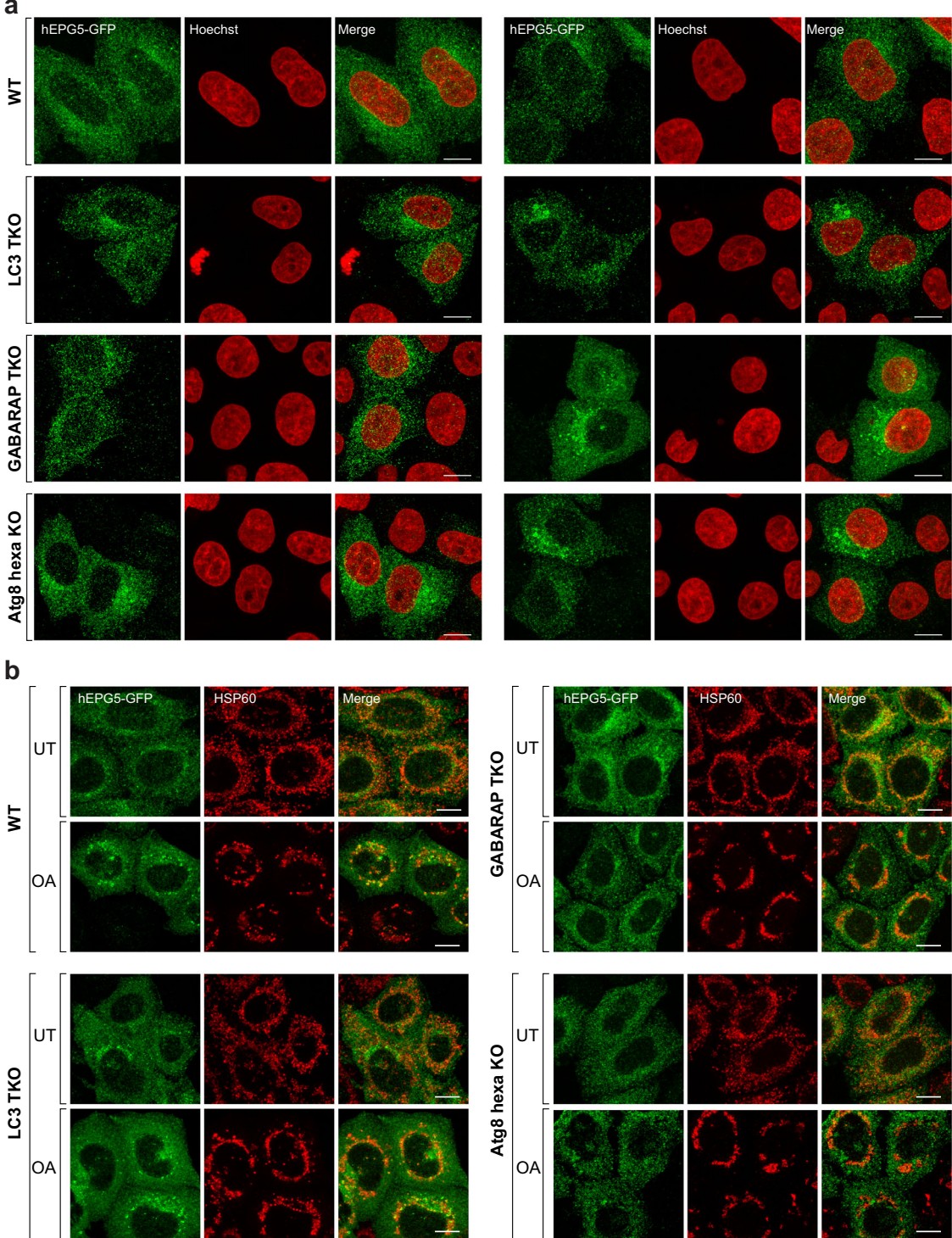

**Fig. 5 hEPG5 localizes to mitochondria in a GABARAP-dependent manner upon activation of Pink1/Parkin-dependent mitophagy. a, b** Wild type (WT), LC3-triple knockout (TKO), GABARAP-TKO, and ATG8-hexaKO expressing mCherry-Parkin were transfected with hEPG5-GFP overnight and subsequently seeded on coverslips to grow for further 48 h. **a** Cells were subjected to immunofluorescence assay (IFA) with anti-GFP antibodies and Hoechst and subsequent imaging by confocal microscope. **b** Cells were left untreated or treated for 3 h with 10 μM oligomycin, 4 μM antimycin A, and 5 μM qVD prior to antibody staining with anti-GFP and anti-mitochondrial HSP60 antibodies and confocal microscopy analysis. Scale bars: 10 μm.

the ability of isolated LIR1 to bind both the noncanonical and canonical sites in the absence of LIR2, validation of this model will likely require high-resolution cryo-EM analysis of full-length hEPG5 in complex with GABARAP or crystallizing GABARAP in complex with the tandem motifs. Finally, our studies here demonstrated that biochemical properties of the isolated LIR peptides may not reflect their true properties in the context of the full-length protein or protein complexes due to factors, such as accessibility.

Based on our EM data, the two LIR motifs are spatially located near the junction point between the hook and the shaft near the center of this protein. This suggested that hEPG5 likely binds

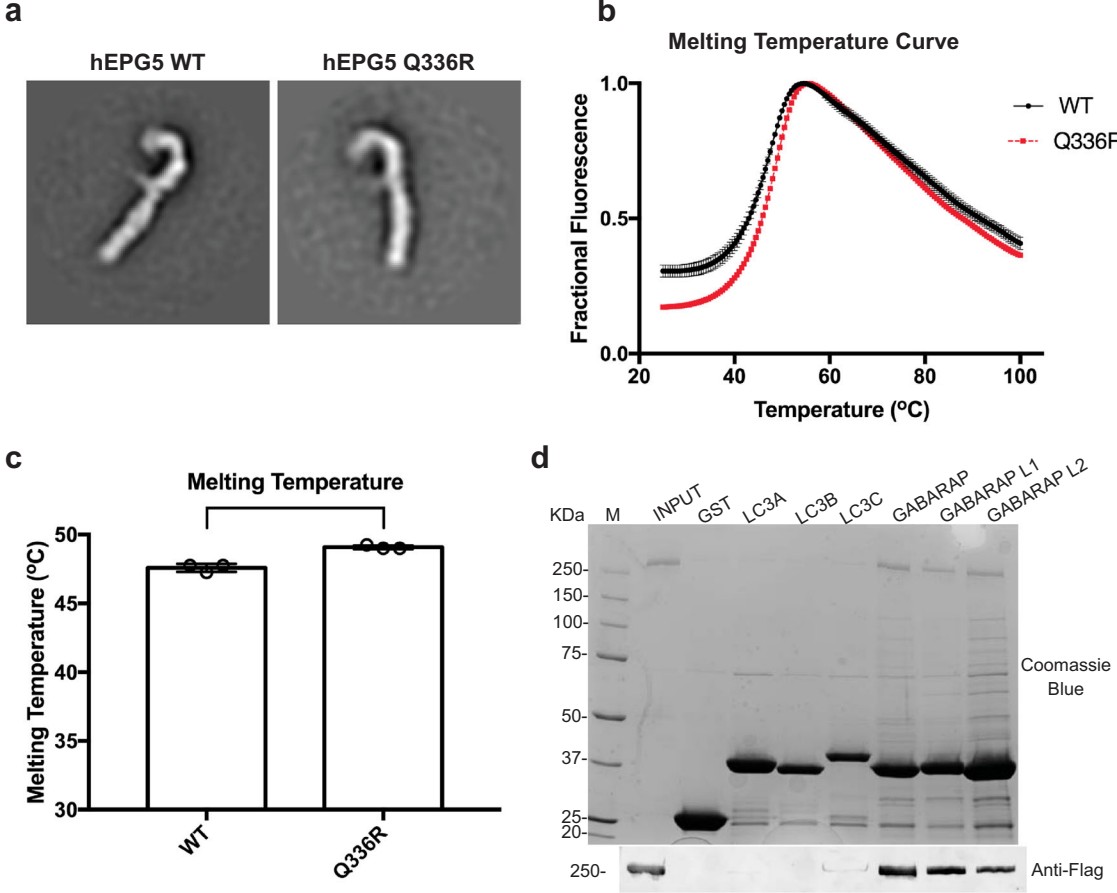

**Fig. 6 Vici syndrome mutant hEPG5$^{Q336R}$ has the same binding specificity and overall architecture as hEPG5$^{WT}$. a** Representative 2D class averages of hEPG5$^{Q336R}$ (right panel) alongside hEPG5$^{WT}$ (left panel). **b** Thermo shift assay of hEPG5$^{WT}$ and hEPG5$^{Q336R}$. hEPG5$^{WT}$ is shown in black and hEPG5$^{Q336R}$ is shown in red. Data are shown as mean ± SEM of three individual experiments. **c** Melting temperatures derived from the first derivative of the melting temperature curve shown in bar graph. Data are shown as mean ± SEM of three individual experiments. **d** In vitro pull-down assays of hEPG5$^{Q336R}$ with GST (control) and GST-tagged LC3/GABARAP subfamily proteins used as baits. Representative SDS–PAGE stained with Coomassie Blue (top panel) shows the hEPG5$^{Q336R}$ and GST constructs input, and representative western blot (bottom panel) was probed by anti-FLAG antibodies. M represents the protein marker. Experiments were performed in triplicates.

autophagosome decorated with GABARAP with its slightly concave shaft facing the surface of the autophagosome. In such a configuration, one could envisage that hEPG5 would tether autophagosome to the lysosome by binding to RAB7 or other lysosomal proteins on its "back" (Fig. 7b). Alternatively, hEPG5 might exert its tethering function by working in conjunction with other factors, such as the HOPS complex at the interface between the autophagosome and lysosome/late endosome[8,62–65].

Although an almost complete catalog of Vici syndrome mutations has been compiled from numerous clinical genetics studies, how these disease mutations affect the structural and biochemical properties of hEPG5 are not known. The system we have built for producing and biochemically and structurally characterizing recombinant hEPG5 can potentially fill a critical gap in investigating the molecular basis of Vici syndrome. We completed a proof-of-concept study by examining hEPG5 encoding the c. 1007 A > G, p. Q336R mutation, which is the most common of four recurrent Vici syndrome mutations reported to date. Patients carrying this mutation show milder symptoms compared to other patients with other mutations, including the lack of cardiac malfunction and immunodeficiency[22,50,52]. Our findings that the Q336R missense mutation does not disrupt the overall architecture, thermal stability, and the GABARAP-binding capability of hEPG5 appear

consistent with these clinical observations. Recent mRNA analysis of a Vici Syndrome patient carrying the c. 1007 A > G, p. Q336R mutation revealed that while alternative splicing caused by this mutation leads to 75% of the transcribed mRNA to contain premature codon truncation and in-frame shift deletion, 25% of the transcribed mRNA are normal spliced product that would lead to the synthesis of full-length hEPG5 protein[50,52]. Further understanding of how the Q336R mutation causes Vici syndrome will require more in-depth investigation of how this mutation affects hEPG5 interaction with RAB7 and the SNARE complex and the autophagosome–lysosome fusion event.

## Methods

**hEPG5 construct cloning, site-directed mutagenesis, and expression**. hEPG5 cDNA, as well as hEPG5 truncation mutants, including hEPG5$^{Δ1–548}$, hEPG5$^{Δ1–1198}$, hEPG5$^{Δ1770–2579}$, and hEPG5$^{Δ2079–2579}$, were cloned into the SalI and NotI sites of a modified pFastBacHTB encoding a His-FLAG tag (gift from Dr. Ji-Joon Song). Site-directed mutagenesis of hEPG5 were carried out using plasmid pFastBacHTB-FLAG-hEPG5 and the following oligos: hEPG5$^{W550A}$, forward 5′-gggtctgggactgcgacgctagtagac-3′ and reverse 5′-gtctactagcgtcgcagtcccagaccc-3′; hEPG5$^{W567A}$, forward 5′-cctgagaccagtgcgattctccttaat-3′ and reverse 5′-attaagga-gaatcgcactggtctcagg-3′; hEPG5$^{F794A}$, forward 5′-gtggacgptgaagacgccataaaaattatt-3′ and reverse 5′-aataattttttatggcgtcttcgtccac-3′; and hEPG5$^{Q336R}$, forward 5′-aaatgtctgtacggggtatctgtgc-3′ and reverse 5′-gcacagatacccgctacagacattt-3′. Addition of EGFP to C-terminus of hEPG5 were carried out using plasmid pcDNA3-FLAG-

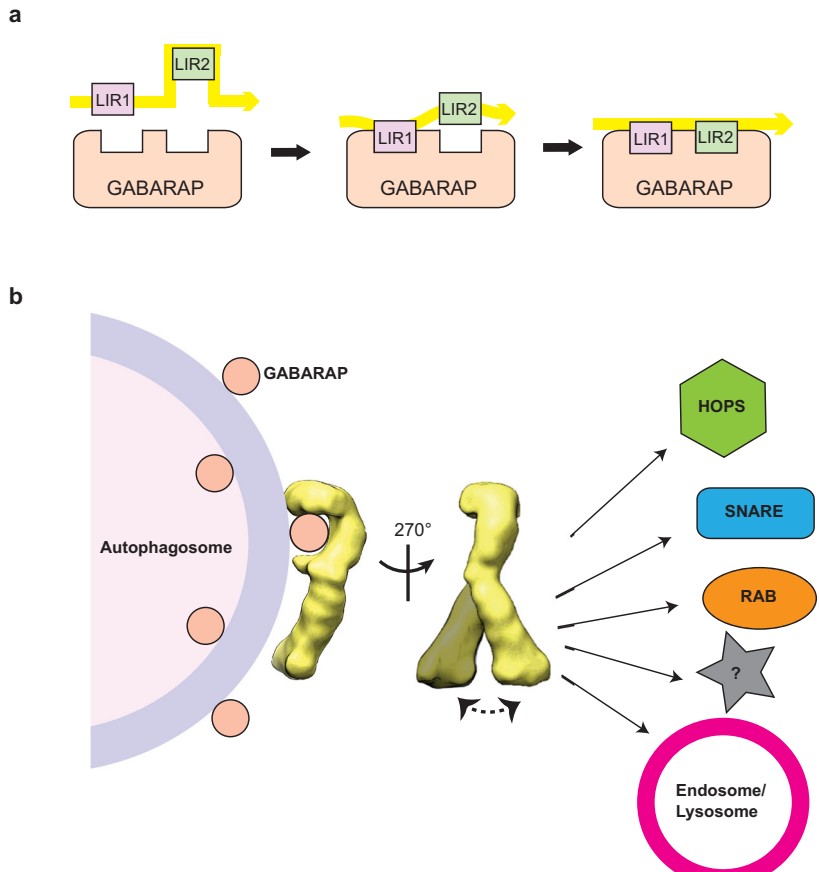

**Fig. 7 Proposed model of hEGP5-mediated tethering in the terminal stage of autophagy. a** Schematic diagram of the hEPG5 tandem LIR motifs binding to GABARAP subfamily in a "two-factor authentication" step-wise binding model. LIR1 is recognized by GABARAP possibly at a noncanonical site initially, with subsequent local conformational change allowing LIR2 becomes accessible to binding at the canonical site on GABARAP. **b** hEPG5 interacts with autophagosome by binding to GABARAP on the concave side of its hook. The convex side of the hook and the flexible stalk may interact with HOPS complex, SNARE proteins, RAB proteins, as well as the late endosome and lysosome to facilitate autophagosome–lysosome fusion.

hEPG5 and the following oligos: forward 5′-agtgcattatttggaccacatacgaccgctcga-gatggtgagcaa-3′ and reverse 5′-ctctagattcgaaagcggccgcctacttgtacagctcgtccatgcc-3′.

cDNA of human ATG8 proteins (LC3A, LC3B, LC3C, GABARAP, GABARAPL1, and GABARAPL2) were synthesized by Thermo Fisher Scientific and subcloned into pQLinkG2, pQLinkH, and pGEX6P-1, using BamHI and NotI restriction enzyme sites. pQLinkG2, pQLinkH, and pGEX6P-1 constructs were expressed for GST pull-down analysis, ITC analysis, and X-ray crystallography, respectively.

**His-FLAG-hEPG5 expression and purification**. Baculovirus containing the wild-type and mutant His-FLAG-hEPG5 constructs, generated using the Baculovirus Expression Vector System, were transfected into Sf9 cells at a density of $1.5–2.2 \times 10^6$ cells/mL. Cells were harvested ~72 h after infection and stored at −70 °C until use.

For purification, Sf9 cell pellets expressing wild-type or mutant His-FLAG-hEPG5 were resuspended in buffer A (50 mM Tris pH 8.0, 150 mM NaCl, 0.1% CHAPS, 1 mM phenylmethylsulfonyl fluoride [PMSF], and cOmplete ethylenediaminetetraacetic acid (EDTA)-free protease inhibitor). The cells were sonicated using Branson Sonicator 450 for four cycles consisting of 20 s sonication followed by 40 s cooling on ice, with duty cycle set to 40% and output control at 4. The resulting cell lysate was centrifuged at $110,200 \times g$ for 30 min at 4°C. The supernatant was then applied to an Anti-FLAG M2 affinity gel (Sigma-Aldrich) for batch binding and eluted with 3× FLAG peptide containing buffer B (50 mM Tris pH 8.0, 150 mM NaCl, and 0.01% CHAPS). The elution containing His-FLAG-EPG5 was applied to the top of a glycerol gradient 15–30% in buffer B using a Gradient station (BioComp, Fredericton). After ultracentrifugation at 38,500 r.p.m. for 17.5 h using Beckman SW55 Ti rotor, fractionation was carried out. The fraction containing His-FLAG-hEPG5 was confirmed by SDS–PAGE gel, subject to preparation of negative stain grids.

**His-FLAG-hEPG5 and GST-GABARAP co-purification**. His-FLAG-hEPG5 was purified using the above protocol but with two different buffers; lysis buffer C (50 mM NaPhosphate pH 7.4, 150 mM NaCl, 0.05% TWEEN 20, 5% glycerol, 1 mM PMSF, and cOmplete EDTA-free protease inhibitor), and elution buffer D with the same components as the lysis buffer C except 0.01% TWEEN 20. The eluate containing His-FLAG-hEPG5 was applied to a HisPur™ Ni-NTA resin (Thermo Scientific) and washed with buffer D. GST-GABARAP in buffer D was applied to the His-FLAG-hEPG5 bound Ni-NTA resin and gently agitated for 30 min. Unbound GST-GABARAP was washed away with buffer D. Following a 35 mM imidazole wash, GST-GABARAP-bound His-FLAG-hEPG5 was eluted with three times 150 mM imidazole in buffer D and two times 250 mM imidazole in buffer D.

**LC3/GABARAP protein expression and purification**. For GST pull-down analysis, N-terminally GST-tagged LC3 and GABARAP proteins were expressed in *Escherichia coli* (T7 Express) cells. The cells were induced with 1 mM isopropyl β-d-1-thiogalacpyranoside [IPTG] for 4 h at 25 °C. The cell pellets were resuspended in 25 mL buffer E (50 mM Tris pH 8.0, 150 mM NaCl, and 1 mM PMSF). Cells were then sonicated for four cycles consisting of 1 min sonication followed by 2 min cooling on ice, with duty cycle set to 50% and output control at 5. Cell lysate was then centrifuged at $20,950 \times g$ for 40 min at 4 °C. Supernatant was incubated for 1 h at 4 °C with glutathione resin (50% slurry, GenScript) pre-equilibrated with buffer E, while gently inverting. Resin was returned to the column and washed with buffer E. Proteins were eluted with 10 mM reduced glutathione (GoldBio) in buffer F (50 mM Tris pH 8.0, 150 mM NaCl). Free glutathione was removed by dialysis in buffer F and protein concentration was measured by spectrophotometry (Nano-Drop, Thermo Scientific). Glycerol was added to a final concentration of 5%, then solutions were aliquoted and stored at −70 °C.

For ITC studies, N-terminally His-tagged LC3 and GABARAP proteins were expressed in *E. coli* (T7 Express) cells. The cells were induced with 1 mM IPTG for 4 h at 25 °C. The cell pellets were resuspended, sonicated, and centrifuged, as described above with buffer G (50 mM Tris pH 7.0, 150 mM NaCl, and 2 mM PMSF). The supernatant was incubated with HisPur™ Ni-NTA resin (Thermo Scientific) pre-equilibrated with buffer G at 4 °C for 1 h. The resin was washed five times with buffer H (50 mM Tris pH 7.0, 150 mM NaCl) and subsequently ten times with buffer H containing 30 mM imidazole. The bound proteins were eluted

with buffer H containing 100 mM imidazole once, then 300 mM imidazole for four times, and 500 mM imidazole once. Eluted proteins were loaded onto a HiPrepQ FF 16/10 column (GE Healthcare) in buffer I (50 mM Tris pH 7.0) and 1% gradient of buffer J (50 mM Tris pH 7.0, 2 M NaCl). Target protein in flow through were collected, concentrated and further purified by size-exclusion chromatography using Sephacryl S-200 column (GE Healthcare) with buffer H.

For crystallography, N-terminally GST-tagged GABARAPL1 was expressed in *E. coli* (T7 Express) cells. The cells were induced with 1 mM IPTG for 4 h at 25 °C. The cell pellets were resuspended, sonicated and centrifuged, as described above with buffer K (40 mM HEPES pH 7.4, 150 mM NaCl, 2 mM PMSF, and 0.1% Triton X-100), followed by centrifugation at 30,966 × $g$ for 40 min. The supernatant was incubated with glutathione resin (GenScript) pre-equilibrated with buffer K at 4 °C for 1 h. The resin was washed six times with buffer L (40 mM HEPES pH 7.4, 150 mM NaCl). On-column GST-tag cleavage with PreScission protease in buffer L with 1 mM dithiothreitol (DTT) and 1 mM EDTA pH 8.0 was performed at room temperature for 2 h. Target protein in flow through and two times washes were collected and further purified by reverse glutathione resin chromatography. Target protein was then concentrated and further purified by size-exclusion chromatography using Sephacryl S-200 column (GE Healthcare), equilibrated and run in buffer F.

**Isothermal titration calorimetry**. LIR1 peptide ([546]GSGTWTLVDEG[556]) was synthesized by LifeTein and LIR2 peptide ([560]DEDPETWILLN[571]) was synthesized by GenScript. ITC experiments of hEPG5-LIR1 peptide titrating into His-LC3/GABARAP proteins were performed using ITC200 (GE Healthcare now Malvern Panalytical), while experiments of hEPG5-LIR2 peptide titrating into His-LC3/GABARAP proteins were performed using MicroCal PEAQ-ITC (Malvern Panalytical). Purified proteins were dialyzed against buffer M (50 mM Tris pH 7.0, 25 mM NaCl) using 6–8 kDa molecular weight cutoff membrane (Spectrum). Synthetic peptides were dissolved in buffer M. A total of 0.6 mM of peptides were titrated into 0.06 mM of His-LC3/GABARAP proteins in 20 steps at 25 °C. Experiments were performed in triplicates. The ITC data were analyzed with one-site binding model using Origin 7.0 (OriginLab). Refer to Supplementary Table 1 for thermodynamic parameters of the ITC experiments.

**Crystallization and data processing**. hEPG5-LIR2 peptide was dissolved in buffer F. GABARAPL1 (10 mg/mL) was incubated with dissolved hEPG5-LIR2 peptide (4.4 mg/mL) at 4 °C for 1 h prior to all crystallization trials. Crystals grew in a condition containing 0.2 M ammonium sulfate, 0.1 M MES pH 5.5, and 29% (w/v) PEG4000 in 1:3 protein to liquid ratio. Crystals were harvested and frozen in liquid nitrogen directly prior to data collection.

X-ray diffraction data were collected on Beamline 5.0.2 at the Advanced Light Source (ALS) and processed using DIALS[66]. The structure was solved by molecular replacement using PHASER[67] with the search model 2R2Q. Model building and refinement were performed using PHENIX[68], CCP4[69], and Coot[70]. Refer to Table 1 for data collection and refinement statistics, and Supplementary Fig. 3a, b for structural figure with probability ellipsoids.

**Negative stain electron microscopy and image processing**. Negative stain specimens were prepared as previously described[71]. In brief, each protein sample from the peak fraction of the glycerol gradient or elution from imidazole was adsorbed on a carbon coated grid and stained with uranyl formate. Micrographs were collected at nominal magnification of 49,000× and a defocus of 1–1.5 μm on a Tecnai Spirit transmission electron microscope (FEI) operating at an accelerating voltage of 120 kV and equipped with a FEI Eagle 4 K charge-coupled device camera. For image processing of the five datasets (hEPG5, hEPG5-MBP, EPG5[Δ2079–2579], hEPG5 in complex with GABARAP, hEPG5[Q336R]; Supplementary Table 2), images were binned twice and particles were subsequently selected using Boxer[72] with a box size of 128 × 128 pixels. 2D classification of the selected particles was then carried out using Relion 1.4[73]. The 3D reconstruction of full-length wild-type hEPG5 was determined using ab initio model function and further refined using cryoSPARC v2 (ref. [74]). A final resolution of 21 Å was calculated using the gold standard method (Supplementary Fig. 4).

**Pull-down assay**. Pulldowns were performed with a 50 μL slurry of glutathione resin (50% slurry, GenScript). The resin was equilibrated with buffer N (50 mM Tris pH 8.0, 150 mM NaCl, 5% glycerol, 0.01% Tween20, and 0.5 mM DTT) and the beads were incubated with equal amounts (200–240 μg) of purified GST, GST-LC3A/B/C, and GST-GABARAP/L1/L2 bait for 10 min at 4 °C with gentle inversion. The tubes were centrifuged at 500 × $g$ for 2 min, followed by removal of excess bait protein in the supernatant. The resin was then incubated with equal amounts (25–30 μg) of purified His-FLAG-hEPG5, truncated hEPG5 or its corresponding mutants (W550A, W567A, W550A/W567A, F794A, or Q336R). The tubes were once again centrifuged at 500 × $g$ for 2 min, followed by removal of the supernatant. The resin was washed with 1 mL buffer N, centrifuged at 500 × $g$ for 2 min, followed by removal of the excess buffer with five washes. After the final wash and removal of the supernatant, 50 μL 2× SDS loading dye was added to each tube. The tubes were then heated at 65 °C for 10 min, loaded onto two 6–15% gradient SDS–PAGE gels, and stained with Coomassie Blue. For western blots, samples were

transferred from a 6–15% gradient SDS–PAGE gel to a nitrocellulose membrane and blocked with Odyssey Blocking Buffer (LI-COR). Mouse anti-FLAG was used as the primary antibody (1:2000; Sigma-Aldrich) and donkey anti-mouse IRDye 680LT as the secondary antibody (1:7500; LI-COR) for visualization on a ChemiDoc Imager (Bio-Rad). Experiments were performed in triplicates.

**Immunofluorescence**. All cell lines were cultured with DMEM containing 10% (v/v) FBS (Cell Sera Australia), 1% penicillin–streptomycin, 25 mM HEPES, Gluta-MAX (Life Technologies), and nonessential amino acids (Life Technologies). Transfection of hEPG5-GFP into these cell lines and WT control was done for 24 h using Lipofectamin 2000 (Life Technologies). Cells were then seeded into six-well plates with HistoGrip (ThermoFisher) coated glass coverslips. After 48 h, cells were left untreated or treated for 3 h with 10 μM oligomycin (Calbiochem), 4 μM antimycin A (Sigma), and 5 μM qVD (MedChemExpress). Cells were then fixed with 4% (w/v) paraformaldehyde in 0.1 M phosphate buffer (10 min), washed three times with 1× PBS, and permeabilized with 0.1% (v/v) Triton X-100 in PBS (10 min). Following a 15 min incubation with blocking buffer containing 3% (v/v) goat serum in 0.1% (v/v) Triton X-100/PBS, samples were incubated with indicated antibodies (anti-GFP (a10262; ThermoFisher); anti-mitochondrial HSP60 (128567; Abcam)) for 1 h, rinsed three times with 1× PBS and incubated with secondary antibodies conjugated to chicken AlexaFluor-488 and/or mouse AlexaFluor-647 (ThermoFisher) for 1 h. The coverslips were rinsed three times with 1× PBS and incubated with 1 μM Hoechst 33342 (ThermoFisher) for 5 min if required before being mounted on glass slides using a TRIS buffered DABCO-glycerol mounting medium.

Images were obtained in 3D by optical sectioning using an inverted Leica SP8 confocal laser scanning microscope equipped with an 63×/1.40 NA objective (oil immersion, HC PLAPO, CS2; Leica microsystems). Imaging was conducted at ambient room temperature using a Leica HyD Hybrid Detector (Leica Microsystems) and the Leica Application Suite X (LASX v2.0.1) with a minimum $z$-stack range of 1.8 μm and a maximum voxel size of 90 nm laterally ($x,y$) and 300 nm axially ($z$). Presentative images are displayed as $z$-stack maximum projections.

**Differential scanning fluorimetry**. Purified wild type His-FLAG-hEPG5 and His-FLAG-hEPG5[Q336R] at 0.4 mg/mL were mixed with 10× Sypro Orange dye in buffer F in a final volume of 25 μL. The fluorescence was measured using the MiniOpticon Real-Time PCR system in triplicates. Melting temperature was calculated using the maximum of the first derivative with Prism 7 (GraphPad).

**Statistics and reproducibility**. All experiments were performed in triplicates. ITC data represents means or means ± SEM. Intensity of His-FLAG-hEPG5, GST (control), and GST-LC3/GABARAP subfamily proteins in pull-down assays was quantified using Bio-Rad Image Lab Software v6.0. Statistics were performed using Prism 7 (GraphPad).

**Reporting summary**. Further information on research design is available in the Nature Research Reporting Summary linked to this article.

## Data availability
The 3D reconstruction of hEPG5 and atomic coordinates for GABARAPL1-hEPG5-LIR2 complex are available at the Electron Microscopy Data Bank (accession code: EMD-23120) and the Protein Data Bank (accession code: 7JHX), respectively. All source data used for generating graphs and charts in main figures are included in Supplementary Data 1. All other data are available from the corresponding author on reasonable request.

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

## Acknowledgements

We thank Omid Haji-Ghassemil and Filip Van Petegem for advice on isothermal titration calorimetry, and Dr. Filip Van Petegem and Dr. Harry Brumer for access to ITC instruments. We thank Franco Li for assistance in crystallographic data collection and processing. Crystallographic data was collected at beamline 5.0.2 at the Advanced Light Source, a U.S. DOE Office of Science User Facility under Contract No. DE-AC02-05CH11231, which is supported in part by the ALS-ENABLE program funded by the National Institutes of Health, National Institute of General Medical Sciences, Grant P30 GM124169-01. Molecular graphics and analyses performed with UCSF Chimera (developed by the Resource for Biocomputing, Visualization, and Informatics at the University of California, San Francisco, with support from NIH P41-GM103311) and PyMOL Molecular Graphics System (Schrödinger, LLC). This work was supported by a CIHR Foundation Grant (FDN-143228) and a CIHR Project Grant (PJT-168907) to C.K. Y. M.L. is supported by NHMRC (APP1160315), ARC (DP200100347), and an ARC Future Fellowship (FT1601100063).

## Author contributions

S.-E.N. and Y.W.S.C. were involved in conception and design, acquisition of structural and biochemical data, analysis and interpretation of data, and drafting or revising the article; M.G. and S.C. were involved in acquisition of biochemical data and analysis and interpretation of data. T.N.N. and M.L. were involved in acquisition, analysis, interpretation of cell imaging data, as well as drafting of the article. C.K.Y. was involved in conception and design, coordinating the project, analysis and interpretation of data, and drafting or revising the article.

## Competing interests

The authors declare no competing interests.
