## [Peer Review File · Communications Biology]

Reviewers' comments:

Reviewer #1 (Remarks to the Author):

Nam et al. investigated the structural and biochemical characterization of human autophagy factor EPG5 for its autophagosome-lysosome tethering activity. The authors showed that hEPG5 adopts an extended shepherd's staff architecture, binds to GABARAP protein, and is critical to autophagosome-lysosome fusion. Further, they showed using mutated proteins to understand the molecular phenotype reported in Vici syndrome. This is an interesting work, and the following are the comments to the authors-

1. Figure 2a, 2c, 2d, and 2e. The authors must mention how many times the experiments were performed.

2. Figure 5. The authors have done the imaging after overnight transfection. Is this time is sufficient to express enough proteins to see the detectable changes? What about the experiment were performed at different time intervals in the different cell lines.

3. Figure 7. What happens if cells are treated with molecules known to induce autophagy and monitor the molecular changes in the cells.

Reviewer #3 (Remarks to the Author):

The manuscript by Dr. Yip and colleagues establishes the overall architecture of the large metazoan tethering protein EPG5, which is known to promote the fusion/degradation of autophagosomes with/in lysosomes. The structure fits with that of previously described tethers, and the authors confirm and characterize in detail the previously reported binding of EPG5 to human Atg8 family proteins (GABARAPs and LC3s) via its two functional LIR motifs. Lastly, the most frequent Vici syndrome-causing mutant version of EPG5 was also analyzed in recombinant form but did not really differ from the wild type protein in stability and Atg8 binding. I do miss many potential additions, such as the analysis of other EPG5 interactions (e.g. with SNAREs and Rab7) and testing (or at least discussing) other Vici-causing point mutant versions, but completing these experiments would take longer than what could be expected for a major revision. Taken together, I think this work could be published in this journal because the results are more-or-less novel, the paper provides strong evidence for its conclusions, the data are technically sound and interesting to the specific sub-field of biology.

Minor comment:

1. I do not agree with the statement "Q336R mutant protein is likely compliant in fulfilling its role in enforcing autophagosome-lysosome/late endosome fusion specificity in autophagy" because neither its autophagosomal/lysosomal recruitment nor its binding to SNAREs and Rab7 were tested. Please either rephrase or include the necessary experiments.

Gabor Juhasz

Reviewer #4 (Remarks to the Author):

EPG5 is a protein necessary for the fusion of autophagosomes to lysosomes thus playing a

fundamental role in the final steps of autophagy.

Mutations of human EPG5 cause a multi-system disorder, known as Vici syndrome and characterized by agenesis of the corpus callosum, cataract, cardiomyopathy, hypopigmentation, and combined immunodeficiency. Different mutations have been described and are associated to diseases of variable severity. The impact on the human mutations on the tethering function of EPG5 has never been demonstrated.

The paper describes a recombinant full length human EPG5 (hEPG5) produced by a baculovirus-insect cell-based system. By negative stain single particle electron microscopy (EM) and 2D analysis, the Authors show that EPG5 is a monomeric elongated molecule with a rigid round "hook" connected to an extended and more flexible "shaft". The tip of the shaft corresponds to the C-terminus of hEPG5. By immunoprecipitation, the Authors demonstrate that hEPG5 preferentially interacts with the GABARAP subfamily of ATG8 proteins via the two LIR motifs located between residues 550 and 570. The paper also shows that mutations preventing the interaction with GABARAPs do not affect the intracellular localization of hEPG5. Finally, the common missense mutation at position 1007 causing the single residue change (Gln336Arg) in hEPG5 protein does not influence its ability to interact with GABARAPs.

The paper is well written and describes interesting data. The methods and tools used appear reliable and solid. I am sure that future studies will be able to identify the molecular mechanisms at the basis of the different forms of Vici syndrome.

We thank the editor and the three reviewers for taking the time to review our manuscript in depth, and providing critical feedback. We have made a number of changes to the manuscript in response to these comments. Please find our responses below. As requested by the editor, we have also included more details of our negative stain EM experiments in the Materials and Methods section as well as in the Supplementary (eg. dataset sizes, FSC curve, etc.).

Reviewer #1

“Nam et al. investigated the structural and biochemical characterization of human autophagy factor EPG5 for its autophagosome-lysosome tethering activity. The authors showed that hEPG5 adopts an extended shepherd's staff architecture, binds to GABARAP protein, and is critical to autophagosome-lysosome fusion. Further, they showed using mutated proteins to understand the molecular phenotype reported in Vici syndrome. This is an interesting work, and the following are the comments to the authors.”

We are grateful for the positive feedback from this reviewer.

Comment 1: *“Figure 2a, 2c, 2d, and 2e. The authors must mention how many times the experiments were performed.”*

Response: We thank this reviewer for pointing out this error. We have included the number of individual experiments for Figure 2 in the figure legend as well as the method and material section.

Comment 2: Figure 5. The authors have done the imaging after overnight transfection. Is this time is sufficient to express enough proteins to see the detectable changes? What about the experiment were performed at different time intervals in the different cell lines.

Response: We are sorry for the lack of detail in the figure legend. As described in the Immunofluorescence section of the revised Materials and Methods, following overnight transfection, the cells were seeded onto glass coverslips and left growing for 48 h prior to immunofluorescence assay and subsequent imaging. Thus, there should be enough time for the protein to be expressed. We have now added this detail into the figure legend of Figure 5.

Comment 3: What happens if cells are treated with molecules known to induce autophagy and monitor the molecular changes in the cells.

Response: According to the reviewer's suggestion, we analysed the localization of EPG5-GFP upon PINK1/Parkin mitophagy induction with Oligomycin A and Antimycin as described previously (Nguyen et al, (2016) *J Cell Biol*). We found that under mitophagy inducing condition, EPG5-GFP localized to structures on and next to mitochondria in WT and LC3 TKO cells (see new Figure 5b), the absence of GABARAPs (GABARAP TKO and Atg8 hexa KO cells) appears to prevent the formation of these structures (new Figure 5b). These results indicate that EPG5 requires GABARAPs for its recruitment to mitochondria during mitophagy, and that EPG5 functions downstream of GABARAPs to drive autophagosome-lysosome fusion.

Reviewer #3

“The manuscript by Dr. Yip and colleagues establishes the overall architecture of the large metazoan tethering protein EPG5, which is known to promote the fusion/degradation of autophagosomes with/in lysosomes. The structure fits with that of previously described tethers, and the authors confirm and characterize in detail the previously reported binding of EPG5 to human Atg8 family proteins (GABARAPs and LC3s) via its two functional LIR motifs. Lastly, the most frequent Vici syndrome-causing mutant version of EPG5 was also analyzed in recombinant form but did not really differ from the wild type protein in stability and Atg8 binding. I do miss many potential additions, such as the analysis of other EPG5 interactions (e.g. with SNAREs and Rab7) and testing (or at least discussing) other Vici-causing point mutant versions, but completing these experiments would take longer than what could be expected for a major revision. Taken together, I think this work could be published in this journal because the results are more-or-less novel, the paper provides strong evidence for its conclusions, the data are technically sound and interesting to the specific sub-field of biology.

I think this work could be published in this journal because the results are more-or-less novel, the paper provides strong evidence for its conclusions, the data are technically sound and interesting to the specific sub-field of biology.”

We are grateful for the positive feedback from this reviewer.

Comment: I do not agree with the statement "Q336R mutant protein is likely compliant in fulfilling its role in enforcing autophagosome-lysosome/late endosome fusion specificity in autophagy" because neither its autophagosomal/lysosomal recruitment nor its binding to SNAREs and Rab7 were tested. Please either rephrase or include the necessary experiments.

Response: We agree with the interpretation from this reviewer and have thus removed this sentence in the revised manuscript. Comprehensive analysis of RAB7 and SNARE binding to wild type and mutant EPG5 is unfortunately beyond the scope of this manuscript. As such, we have included these suggested experiments as future directions in the concluding paragraph of the revised manuscript.

Reviewer #4

“EPG5 is a protein necessary for the fusion of autophagosomes to lysosomes thus playing a fundamental role in the final steps of autophagy.

Mutations of human EPG5 cause a multi-system disorder, known as Vici syndrome and characterized by agenesis of the corpus callosum, cataract, cardiomyopathy, hypopigmentation, and combined immunodeficiency. Different mutations have been described and are associated to diseases of variable severity. The impact on the human mutations on the tethering function of EPG5 has never been demonstrated.

The paper describes a recombinant full length human EPG5 (hEPG5) produced by a baculovirus-insect cell-based system. By negative stain single particle electron microscopy (EM) and 2D analysis, the Authors show that EPG5 is a monomeric elongated molecule with a rigid round “hook” connected to an extended and more flexible “shaft”. The tip of the shaft corresponds to the C-terminus of hEPG5. By immunoprecipitation, the Authors demonstrate

that hEPG5 preferentially interacts with the GABARAP subfamily of ATG8 proteins via the two LIR motifs located between residues 550 and 570. The paper also shows that mutations preventing the interaction with GABARAPs do not affect the intracellular localization of hEPG5. Finally, the common missense mutation at position 1007 causing the single residue change (Gln336Arg) in hEPG5 protein does not influence its ability to interact with GABARAPs.

The paper is well written and describes interesting data. The methods and tools used appear reliable and solid. I am sure that future studies will be able to identify the molecular mechanisms at the basis of the different forms of Vici syndrome.”

We are grateful for the positive feedback from this reviewer.

REVIEWERS' COMMENTS:

Reviewer #1 (Remarks to the Author):

The authors satisfactorily answered my comments and performed new experiments and modified the manuscript accordingly.

Reviewer #3 (Remarks to the Author):

I think that this paper can now be published.